# Dual Effects of N-Butanol and Magnetite Nanoparticle to Biodiesel-Diesel Fuel Blends as Additives on Emission Pattern and Performance of a Diesel Engine with ANN Validation

**Ahmed Sule** [1,2,*]**, Zulkarnain Abdul Latiff** [1,*]**, Mohd Azman Abas** [1]**, Ibham Veza** [3]**,
Manzoore Elahi M. Soudagar** [4,5]**, Irianto Harny** [6] **and Vorathin Epin** [3]

[1] Automotive Development Centre, School of Mechanical Engineering, Faculty of Engineering, Universiti Teknologi Malaysia, Johor Bahru 81310, Malaysia

[2] Automotive and Mechanical Technology Education Section, Technical Education Department, Kogi State College of Education, Ankpa 1033, Nigeria

[3] Department of Mechanical Engineering, Universiti Teknologi PETRONAS, Perak 32610, Malaysia

[4] Department of Mechanical Engineering, University Centre for Research and Development, Chandigarh University, Mohali 40413, India

[5] Department of Mechanical Engineering, School of Technology, Glocal University, Delhi-Yamunotri Marg, Mirzapur Pole 247121, India

[6] Department General Education, Faculty of Resilence, Rabdan Academy, Abu Dhabi P.O. Box 114646, United Arab Emirates

[*] Correspondence: author: ahmedsule@graduate.utm.my (A.S.); zkarnain@utm.my (Z.A.L.)

**Abstract:** This paper investigates impact of magnetite dispersed in butanol and added to two varied blends of palm biodiesel and diesel (B20 and B30). The developed fuel samples were characterized and tested on single cylinder diesel Yanmar engine (L70N) to observe engine behavior for emissions and performance. Results are compared with two reference fuels: YF50 fuel contains 50 ppm magnetite in B20 and $B_n10Y90$ contains 10% butanol with 90% B20. Addition of magnetite and butanol depletes emissions levels and improve performance compared to ordinary B20 and B30 however; samples with higher dosage of magnetite (150 ppm) yielded better results in performance and emission compared with lower dosage (75 ppm). The best sample was C10Z90 which entails 150 ppm magnetite in butanol added at 10% to B30. Brake thermal efficiency (BTE) at highest brake power (BP) point for C10Z90 was 37.28% compared to others (32.88%, 35.22% and 35.96%). Additionally, brake specific fuel consumption (BSFC) of C10Z90 was at least 8.29 g/Kw.hr and at most 84.52 g/Kw.hr less than other samples at highest BP point. Results indicated C10Z90 was lower in carbonmonoxide, hydrocarbon and smoke except for oxides of nitrogen. Artificial Neural Network (ANN) model successfully predicted BTE, BSFC and emissions of the dual fuel application.

**Keywords:** magnetite; emissions; ANN model; efficiency; butanol

## 1. Introduction

The population of the world according to world bank have grown with over 1 billion between year 2010 and 2021 with corresponding urbanization of rural settlements which drives the needs for higher social life and mobility. As a result, the demand for automotive vehicles for mobility increases with these trends and consequently, higher combustion of fossil fuels for the vehicles results. The increase in the combustion of fossil fuel further causes increment in emission of gases which facilitates climate change and global warming. Many governments therefore created legislations to regulates the automotive industries by adoption of minimum standard for gases emissions without compromising efficiency while maintaining reasonable level for rate of fuel consumption. As a result, three

methods have been adopted to improve performance of engine with respect to efficiency of energy utilization and emission reductions viz: engine combustion system modifications [1]; use of renewable and green energy fuels like biofuels and biodiesel [2,3]; and exhaust gas treatment [4]. Notably, Carbon oxide (CO) and Hydrocarbon (HC) emissions are higher when using conventional diesel compared to biodiesel from consideration of biodiesel characteristics in terms of high flash points, improved inherent lubricity, biodegradability and non-toxic nature, although nitrogen oxide (NOx) remains high mostly at higher engine loads with biodiesel use. Previously, the means of reducing NOx was by adjustment of timing for fuel injection. Nowadays, in order to achieve more desirable results biodiesel is been blended with conventional diesel and the limitations with use of only conventional diesel recorded to be minimal but for further improvements on the results obtained from blending of biodiesel and diesel at different ratios, the use of additives have become promising [5].

The use of oxygenated fuels like methanol, ethanol or n-butanol as additives with only conventional diesel has been experimented and the findings showed improvement in terms of emission reduction and small impact on engine performance [6,7], other findings also reported similar results when the oxygenated fuels are blended with biodiesels alone [8,9]. Furthermore, recent investigations have showed that the applications of nanoparticles have significant effects on many engineering systems and as a result, been used singly or as additives to other components or composites materials including for enhancement of heat transfer [10], friction and wear reduction [11], optical usage [12], industrial cooling [13] and as nano-additives or nano-catalyst to biodiesel-diesel blends or separately to conventional diesel or biodiesel among other applications. Many categories (majorly metals and their oxides) of nano-catalyst or nano-additives been used with diesel-biodiesel fuels have been reported to exhibit considerable influence on exhaust gas emission and engine performance such as cerium oxide ($CeO_2$) [14–16], aluminum oxide ($Al_2O_3$) [17–20], carbon nanotube (CNT) [21–23], Zinc oxide (ZnO) [24,25] and titanium oxide (TiO) [26,27] among others and novel ones still been investigated. The ability of the nano-additives to influence these properties is attributed to their high surface to volume ratio as well as their inherent energy carrying capacity. The nano-additive changes specific chemical and physical properties in the fuel, some of the variations with the diesel-biodiesel fuels are in terms of fuel flash points, lower and higher heating values, kinematic or dynamic viscosity, density and cetane number which thus leads to reported results in terms of time reduction in ignition delay (ID), increase in break thermal efficiency (BTE), better fuel utilization noticeable from the break specific fuel consumption (BSFC), power output (PO) increase at higher loads compared to only diesel fuel usage, overall emission reductions specifically CO, NOx and HC.

Nano-additives applications has many challenges which hinders the research, some of the challenges are temperature instability possibly resulting from particle cohesion from the Brownian motion, gravitational effects causing settling or sedimentation of the nanoparticles in the base fluids, coagulation formation due Vader Waal forces leading to interfacial layering and lastly high cost of nanoparticles due to advanced technical and professional requirement for their synthesis, but a major drawback is in terms of stability in the base fluid after blending, to tackle this issue, the use of surfactant has been reported to be essential [,28]. However, some surfactants have effects on the potency of the nanoparticle catalyst when used for stabilization and for this reason this work target using ultrasonication with butanol. N-butanol as whole fuel or as additive has many advantage over ethanol and acetone [3], challenges with ethanol as biofuel ranges from its lower density and hygroscopicity compared to conventional diesel while butanol as a higher alcohol exhibits higher heating value, more efficient solubility and also lower corrosivity making it cleaner, efficient and affordable as fuel or additives in combustion engines although its main limitations is with regards to difficulty in mass production but recently many improved techniques for mass producing butanol have been identified [7]. The use of iron oxide nanoparticle as additive in diesel engine have not been reported among

recent research; considering that using nano-additives improves engine performance and reduce emission as similar to butanol use as additives, this have encouraged the idea of this research paper to investigate the combined impact of both fuel additives (dual additives) at varying percentage ratio. Furthermore, in this study, the fuel blends characterization was performed using the applicable American standard for testing and materials (ASTM) method and then artificial neural network (ANN) was adopted in modelling the data from the performed experiments and predicts most feasible optimized fuel blend that best fit in terms of performance and emissions among the selected fuel blends consequently eliminating excess cost that may arise from testing more varieties of the nano-additives.

## 2. Experimental Materials and Methods

In this investigative research some of the required materials and samples were purchased while others were produced by the authors of this work. Details of this variations in how the research materials as well as how the fuel samples were sought in addition to processing the blends is provided in the following sub-sessions.

### 2.1. Materials Sources

The pure biodiesel termed B100 was supplied by the Malaysian Palm Oil Board (MPOB) industry through a research request by the Automotive Development Centre (ADC) of the Universiti Teknologi Malaysia (UTM). According to manufacturer's product statement and description, the esterification process had been used in producing the biodiesel from palm oil source in order to yield the palm oil methyl ester (POME). On the other hand, pure conventional diesel in this work termed ($D_z$) was specially requested and purchased from *Petronas company* Malaysia (an oil and gas production and exploration company with retail outlets). Iron Oxide was synthesized at the chemical laboratory (Nanotechnology and Nanomaterials sections) of the Universiti Teknologi Malaysia.

### 2.2. Methods for Processing Blends and Experimentation

The coprecipitation technique was adopted for the processing and production of the iron II Oxide nano-additive. An iron sulphate $FeSO_4$ ($Fe^{2+}$) salt solution and an iron chloride $FeCl_3$ ($Fe^{3+}$) solution and ammonium hydroxide $NH_4OH$ solution were set on the chemical production table. The molar ratio of $FeCl_3$ ($Fe^{3+}$) to $FeSO_4$ ($Fe^{2+}$) was one (1) to two (2). An electric powered magnetic stirrer was used in stirring the solution. Both the $FeSO_4$ ($Fe^{2+}$) and $FeCl_3$ ($Fe^{3+}$) solutions were used as the precursors while $NH_4OH$ solution was utilized as the reducing agent.

In order to have best resulting size of the nanoparticle, the impact of the reducing agent on the final size of nanoparticle was considered and therefore optimal volume of the reducing agent was used. The $FeSO_4$ ($Fe^{2+}$) was added to the $FeCl_3$ ($Fe^{3+}$) and the mixture was continuously stirred for 15 min using the RCT 5 digital magnetic stirrer 850 W at selected speed of 1000 rpm (the device speed range is 50 rpm–1500 rpm). The Iron Oxide nano-additive production for this research work is shown in the flowchart Figure 1 while the governing chemical equation is a presented in Equation (1) to Equation (3). After obtaining a homogenous blend of the initial solution mixture then the stirrer speed was reduced to 900 rpm and $NH_4OH$ solution is then added in dropwise flowrate continuously until a pH of 10 was achieved while further stirring for 22 min, the percentage by volume of $NH_4OH$ solution is about 10% of total mixture. As stirring continues, a black-like precipitate begins to form on the surface. A final cleaning procedure was done as the resulting nanoparticle was tested using a magnet which showed attraction characteristic with the nanoparticle, the particle size of the iron nanoparticle was measured using the Zetasizer Nano (ZS-UK model) of Dynamic Light Scattering (DLS) machine while transmission electron microscopy (TEM) was utilized in viewing the images.

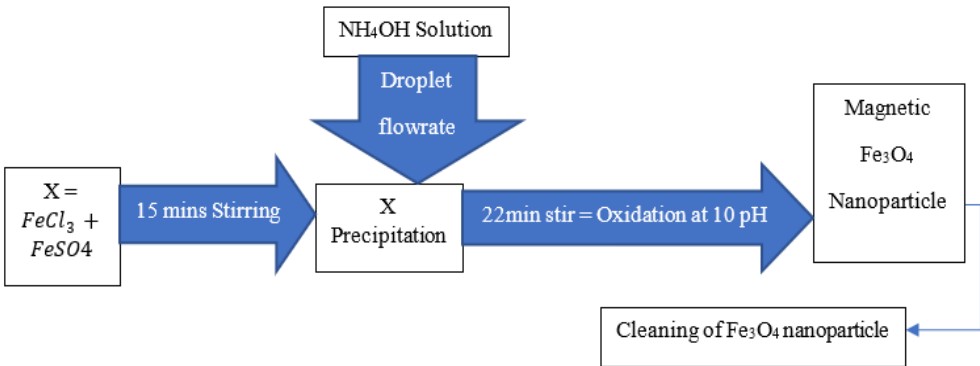

**Figure 1.** Synthesis sequence adopted to produce Iron Oxide nanoparticles.

$$FeCl_3 + 3NH_4OH \rightarrow 3NH_4Cl + Fe(OH)_3 \qquad (1)$$

$$FeSO_4 + 2NH_4OH \rightarrow 3NH_4Cl + Fe(OH)_2 \qquad (2)$$

$$Fe^{2+} + Fe^{3+} + 8OH^- \rightarrow Fe_3O_4 + 4H_2O \qquad (3)$$

*2.3. Fuel Blending*

The fuel samples were prepared at the chemical and fuel section of ADC UTM, the butanol was arbitrarily assigned nomenclature as $B_n$ and was first blended with $75 ppm\ Fe_3O_4$ nanoparticle and then $150 ppm\ Fe_3O_4$ nanoparticle by mass fractions and with the aid of the ultrasonication process, after 20 min a homogenous mixture resulted. On the other hand, for ease in comparison, the general nomenclature of B20 and B30 was change; the pure diesel in this work termed 'D$_z$' and POME was blended at a ratio of 80% to 20% respectively and this blend was referred to as Y. Also, 70% of D$_z$ and 30% POME was blended and termed Z in similar manner. The percentage composition by ratio for each sample and the assigned nomenclature is as presented in Table 1. The final fuel sample was obtained by blending the first stage and the second stage.

**Table 1.** Fuel samples preparation and nomenclature.

| | Sample | Fuel Blend Ratios |
|---|---|---|
| First stage blending: | A = BnF75 | 75 ppm $Fe_3O_4$ + Butanol |
| $Fe_3O_4$ + Butanol | C = BnF150 | 150 ppm $Fe_3O_4$ + Butanol |
| Second stage blending: | Y = B20 | 80% DZ + 20% POME |
| Diesel + Biodiesel | Z = B30 | 70% DZ + 30% POME |
| | A5Y95 | 5% of A + 95% of Y |
| Third stage blending: | A5Z95 | 5% of A + 95% of Z |
| First stage + Second stage | C10Y90 | 10% of C + 90% of Y |
| | C10Z90 | 10% of C + 90% of Z |

The fuel characterization was done under controlled humidity and temperatures in the chemical laboratory to ensure better accuracy of results. In order to be able to compare variations in results, Y was doped with 50 ppm $Fe_3O_4$ directly, and also $B_n$ blended at 10% to 90% of Y. These samples were termed YF50 and $B_n$10Y90 respectively. The results were adopted as training data for the ANN model with respect to prediction and estimation of direct effect of $Fe_3O_4$ on Y and also $B_n$ direct effect on Y as single independent additives as well as correlating this aspect with their impact as dual additives.

*2.4. Engine Set-Up and Specification*

The experimental engine set-up and dynamometer controller is as shown in Figure 2a,b respectively, the engine was linked to the 30-kW dynamometer (eddy current Magtrol model) which was used to vary the engine load from the control panel while running on a constant speed of 2500 rpm. In order to stabilized the speed at 2500 rpm, air fuel ratio (AFR) was set to correlate with the values of brake mean effective pressure (BMEP), a high frequency piezoelectric pressure transducer (of Kistler 601A model) installed on the engine head was used to acquire the in-cylinder pressure at a crank angle degree resolution of 0.2°, also, crank angle encoder (Kistler 2613B model) installed on the crankshaft was utilized in measuring the crank angle degree.

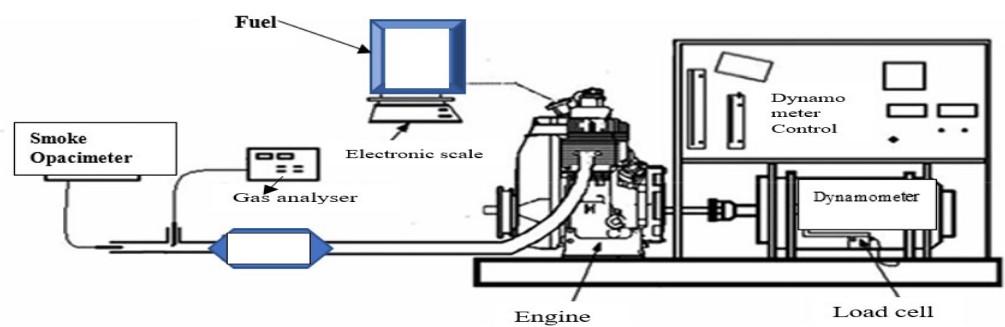

(**a**)

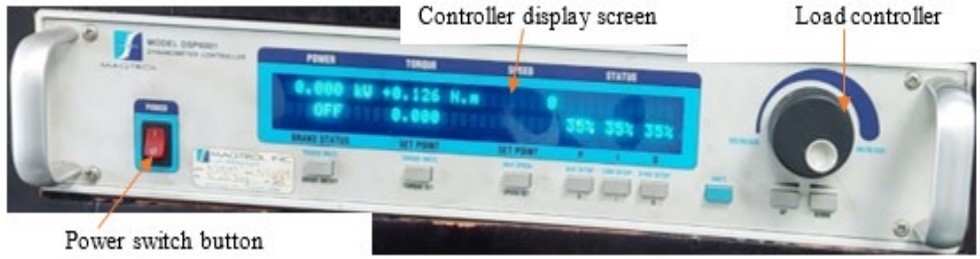

(**b**)

**Figure 2.** (**a**) Schematic diagram of experimental set-up. (**b**) Dynamometer Controller (Magtrol Model DSP6001).

The intake and exhaust temperature were measured using a thermocouple (K-type, accuracy of ± 1.5 °C) and the throttle was opened partially for fuel delivery at 8 g/s maximum on steady run. The variations in engine loads necessitates changes in the air and fuel amount. Specifications of the test engine and exhaust gas analyzer is presented in Table 2 and Table 3 respectively. Initially before starting the experiment, all the data loggers, sensors and equipment were calibrated to achieve accurate data while applying the necessary American Standard for Testing and Materials (ASTM).

**Table 2.** Test engine Specifications.

| Engine Parameter | Specification |
| --- | --- |
| Engine Type | Yanmar L70N |
| Engine geometry | 4 stroke/single cylinder |
| Length of connecting rod (mm) | 102 |
| Stroke/bore (mm) | 67/78 |
| Cooling | Air cooled |
| Compression ratio (CR) | 20:1 |
| Displacement (liters) | 0.320 |

| | |
|---|---|
| Maximum rated output (kW) | 4.9 (at 3600 rpm) |
| Fuel tank capacity (liters) | 2.7 |
| Starting system | Electric/Recoil start |
| Lubrication system | Trochoid pump forced lubrication |
| Length × width × Height | 378 × 422 × 453 |

In addition to the smoke opacimeter, particulate matter was estimated using the vacuum pump to streamline the exhaust gas such that the exhaust gas particle or unburnt by-products is dissipated on the filter paper with pore diameter maximum and minimum values of 10μm and 0.015μm respectively, the scale of dissipation is therefore used to estimate particulate which is then correlated with the smoke opacimeter data. As shown in Figure 3, using the card (SPECTRUM, model MI3112CA) for data acquisition to convert data from analogue to digital which was mount on the DEWE-5000 data acquisition system. The experimental data was then retrieved by applying software developed by *Dewetron.* DEWESoft and DEWECa were used for non-crank angle dependent data and crank angle dependent data respectively. The Technical features of the opacimeter is as shown in Table 4.

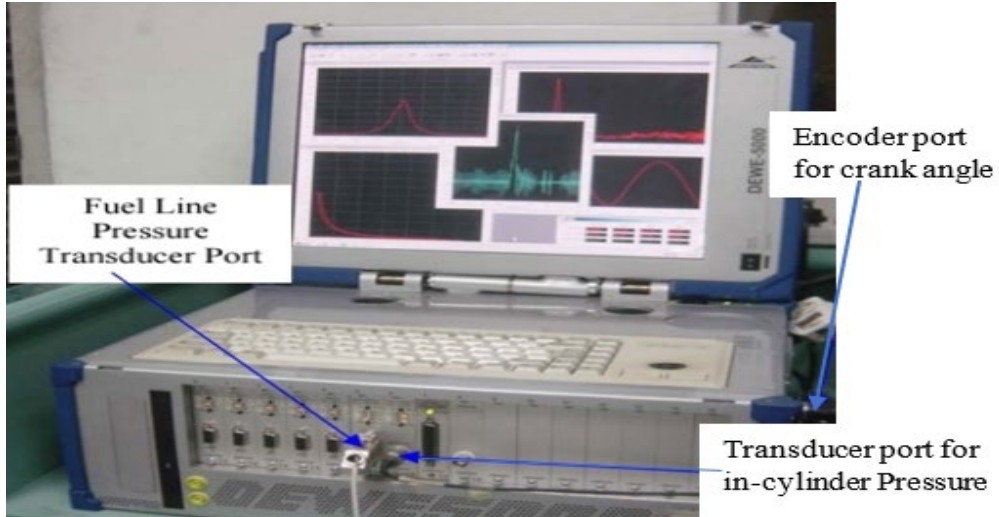

**Figure 3.** Data acquisition (DEWE-5000) system.

**Table 3.** Specifications of the exhaust gas Analyzer (Testo 350XL emission measuring device).

| By-Products of Combustion | Measuring Range | Accuracy |
|---|---|---|
| Carbon dioxide ($CO_2$) | 0–50% vol. | ±0.3% vol.+1%mv ppm |
| Carbon monoxide (CO) | 0–1000 ppm | 5 ppm (0–99 ppm) |
| Oxygen ($O_2$) | 0–25% vol. | ±0.2 mv |
| Nitrogen oxide ($NO_x$) | 0–3000 ppm | 5 ppm (0–99 ppm) |
| Hydrocarbon (HC) | 0.01–4% vol. | <400 ppm (100–4000 ppm) |

**Table 4.** Opacimeter Technical features.

| Parameter | Range in Measurement | Accuracy |
|---|---|---|
| Exhaust smoke density | 0–99% | 0.01 |
| K smoke factor (1/m) | 0–10% | 0.01 |
| Engine speed | 0–9999 (1/m) | 1/m |

The fuel characterization was done under controlled humidity and temperatures in the chemical laboratory to ensure better accuracy of results. The engine was allowed to

run for 10 min to warm up before testing the fuel blends and then the fuel filter was replaced when a different blend sample was to be tested.

### *2.5. Engine Operating Conditions for Experiment*

In this investigation, the experiment was performed on the Yanmar L70N at five different load conditions with the aid of the dynamometer controller; these are: 5% load, 25% load, 50% load, 75% and 100% load while maintaining a steady speed of 2500 rpm. The pure conventional diesel was first used to run the engine for 20 min at steady state conditions then the experiment was performed by altering the load condition while noting the readings of the target parameters in each case before subsequently replacing the conventional pure diesel with fuel samples produced for the investigations in this research starting with A5Y95 sample. In each case for each engine load tested, the experiment was rerun twice to avoid data acquisition errors while taking readings. Subsequently, variation in engine loads necessitates changes in the air and fuel amount. The specific target of this investigation is to discover the impact on engine in terms of behavior changes of $Fe_3O_4$ nano-additive proportion doped in butanol and added as dual fuel to POME-diesel. The achieved results are then optimized, graphically presented and compared and contrasted with a reference fuel with only butanol content with POME-diesel fuel. The collected data was then used as the basis for the ANN model.

### 3. Artificial Neural Network Modelling (ANN Model)

The issues with nano-additive usage as catalyst is temperature instability partly resulting from Brownian motion, viscosity increase corresponding with density, instability or formation of agglomerates and sedimentation due to factors as gravitational forces, coagulation formation in form of lumps due to interfacial layering, and high cost in synthesis of nano-additive [29,30]. As a result of these issues and specifically cost, modelling is necessary and artificial neural network (ANN) is mostly feasible due to its effectiveness compared to other modelling tool as suggested in [31]. The MATLAB 9.11 version R2021b software licensed under the Universiti Teknologi Malaysia therefore was used in developing the predictive model. An ANN tool was selected and the learning algorithm was trained using the feed forward back propagation algorithm as presented in Figure 4.

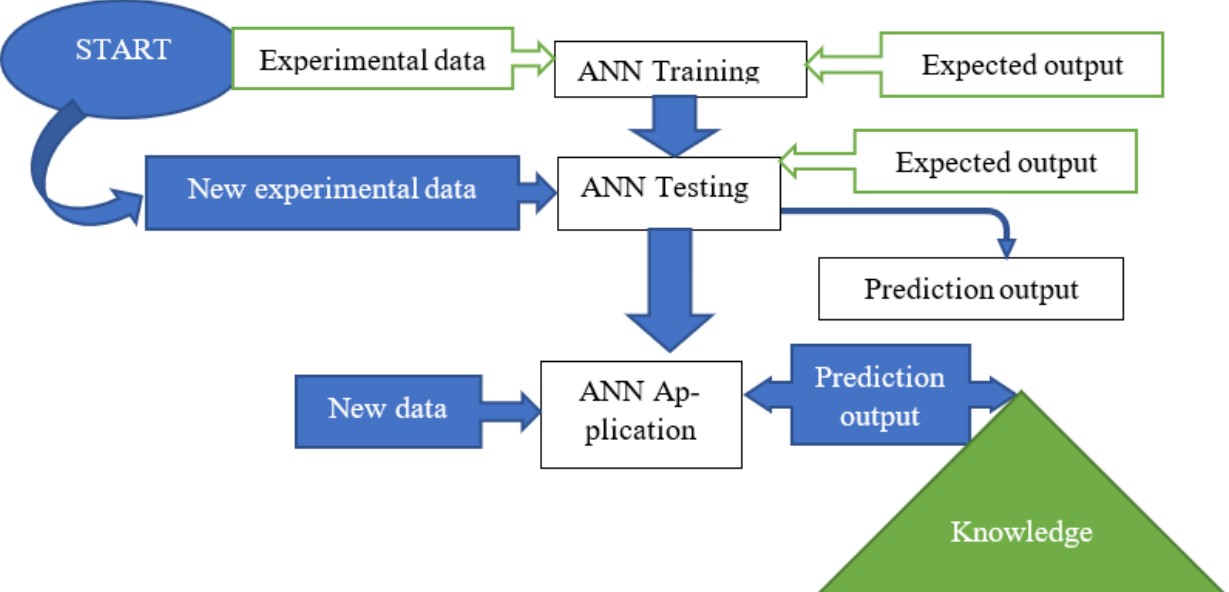

**Figure 4.** ANN working principles.

A random arrangement of data for training, testing and validation was arbitrarily set at 60%, 20% and 20% respectively. Using Equation (4) and Equation (5) denoting correlation coefficient (R) and determination coefficient (R²) respectively, the accuracy of ANN prediction model was examine since previous studies have validated this technique [32].

$$\text{Correlation Coefficient (R)} = \sqrt{1 - \{(\textstyle\sum_{i=1}^{n} (M_i - P_i)^2 \mid \sum_{i=1}^{n} P_i^2\}}} \qquad (4)$$

$$\text{Determination Coefficient } (R^2) = 1 - \{\textstyle\sum_{i=1}^{n} (M_i - P_i)^2 \mid \sum_{i=1}^{n} P_i^2\} \qquad (5)$$

## 4. Results of Experiments and Discussions

The emission pattern and performance of a diesel engine running on POME and conventional diesel blended with alcohol-based fuel (butanol) with $Fe_3O_4$ nano-additive dispersed in it to improved its energy level is discussed. However, in order to examine the results in this study and its impact on general fuel technology, the pre-combustion stage analysis is relevant therefore the micro structure of the $Fe_3O_4$ nano-additive was first examined using two separate magnification of the TEM image as presented in Figure 5. With the estimated particle size.

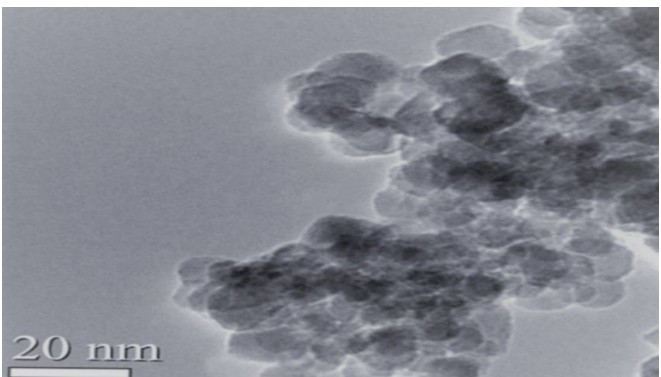

**Figure 5.** TEM image with double magnification of Fe₃O₄ (Resolution increased for clarity).

### 4.1. Particle Size Impact

The mean particle size of the $Fe_3O_4$ doped with butanol was 20 nm approximately, the distribution of this particle size with respect to its frequency is as presented in Figure 6. As a result of this size and from the morphologies of the $Fe_3O_4$ nano-additive, apart from increasing the energy content of butanol it also caused faster ignition and ease the combustion process as similar to findings in [33] which reported that reduction in boron nanoparticles size resulted in ignition at a lower temperature compared to those whose particle sizes were higher. Additionally, as a result of particles sizes, the stability of $Fe_3O_4$ in the fuel samples were noticeably longer in time duration as seen in Figure 7 whereby from day 1 up to 14 days quiescence the prepared fuel samples was clearly stable as agglomeration of the samples was not very visible for the samples YF50, A5Y95, A5Z95 C10Y90 and C10Z90 however, after three weeks of quiescence (21 days), the samples exhibited clear sedimentation that was visible. This can be attributed to biodiesel ability to homogenously align with nanoparticles compared to pure diesel due to nanoparticles tendency to exhibit more hydrophilic behavior with biodiesel compared to pure conventional diesel as this also tally with report from [20] whereby aluminum coated with carbon nanoparticle was relatively stable without sedimentation or formation of agglomerates for specified period of 15 days.

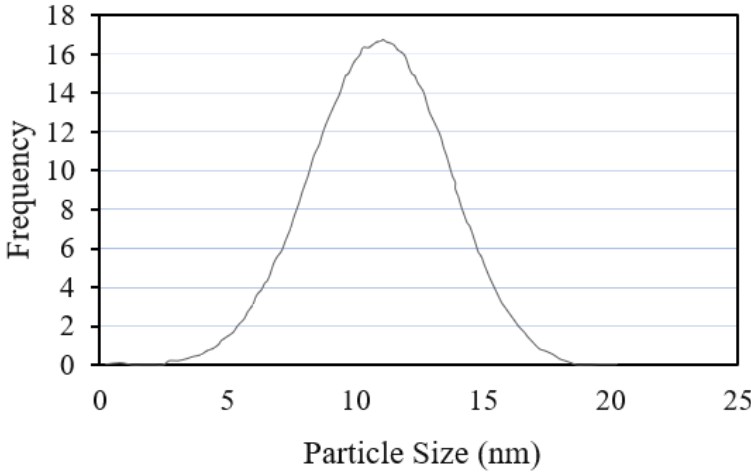

**Figure 6.** Particle size Distribution.

The nano-additive particle size is therefore a contributing factor to the homogenous stability of the samples due to lesser weight per unit volume and lower interfacial layering of the $Fe_3O_4$ nanoparticles.

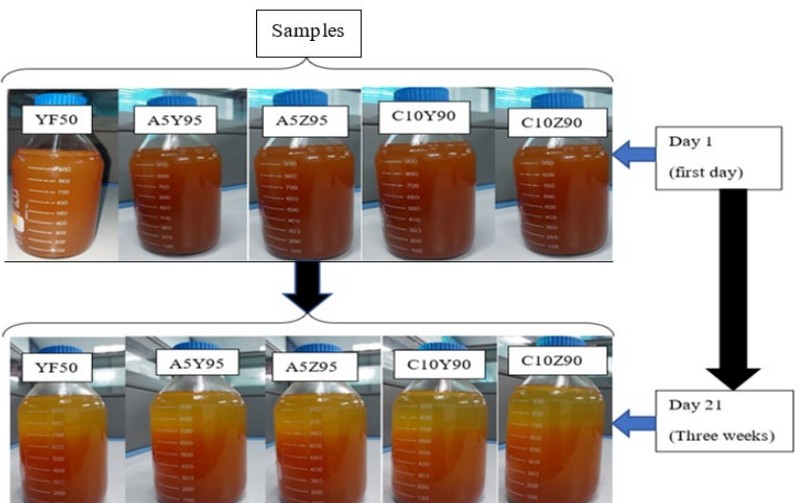

**Figure 7.** Sedimentation duration and profile for Fuel Samples.

*4.2. Characterisation of Fuel Samples*

The fuel blends were characterized by using applicable ASTM standards to estimate the heating values, cetane number, viscosity, density and Acid values. This is presented in Table 5.

**Table 5.** Physicochemical Characterization of Test Fuels.

| Properties | Y (B20) | Z (B30) | A5Y95 | A5Z95 | C10Y90 | C10Z90 | Method |
|---|---|---|---|---|---|---|---|
| Cetane number | 49.56 | 50.65 | 52.06 | 52.20 | 53.60 | 53.08 | ASTMD613 |
| Viscosity (mm²/s) at 40 °C | 3.91 | 3.95 | 3.98 | 4.01 | 4.09 | 4.09 | ASTM 445 |
| Density (kg/m³) at 15 °C | 853.08 | 857.14 | 859.05 | 859.88 | 860.70 | 861.04 | ASTM D4052 |
| Acid value (mgKOH/g) | 0.22 | 0.26 | 0.29 | 0.30 | 0.32 | 0.31 | ASTM D941 |

### 4.3. Performance Parameters

The performance in terms of rate of fuel usage and the thermal efficiency of the engine as well as the exhaust gas temperature is thus discussed in this section. The variations in the prepared fuel samples is thus compared and analyzed.

#### 4.3.1. Brake Thermal Efficiency (BTE)

The BTE is an important feature in the capability of diesel engines which denotes the fractional conversion of chemical energy in the fuel to useful power output. The Figure 8 present the relationship between the BP and the BTE with respect to varying load conditions. It is clear that for all the fuel doped with a combination of butanol and $Fe_3O_4$ nano-additive, the BTE increases with corresponding increase in applied engine loads which can be attributed to decrease in rate of heat rejection with correlation to increasing fuel rate [27,34], as for YF50 sample which exhibited BTE at 27.74% for highest point of load application corresponding to the highest BP released, the BTE is yet lower compared to the other samples A5Z95, A5Y95, C10Z90 and C10Y90 (BTE 32.88%, 35.22%, 37.28% and 35.96% respectively). This of course provides the inference that POME doped with $Fe_3O_4$ nano-additive alone despite been better than only POME as fuel still score lower in BTE when compared with a combination of $Fe_3O_4$ nano-additive and $B_n$ in POME. The reason can thus be assigned to the fact that atomization as well as the spray tendency is lower for only POME and also when blended with any nano-additive alone compared to additional application of alcohol biofuels especially butanol due to its physico-chemical characteristics mainly good solubility, lesser corrosivity and high heating value in comparison with ethanol counterpart [3,35].

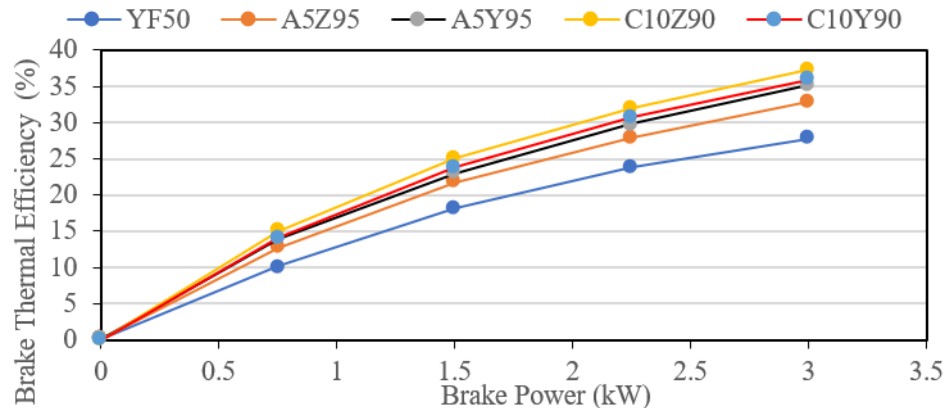

**Figure 8.** BTE Variation with BP.

Furthermore, the application of $Fe_3O_4$ nano-additive in conjunction with butanol created better catalytic stimulation which of course influenced and reduced the ignition delay and enhanced the combustion process to occur more rapidly, the totality of this in turn improves overall BTE however, when critically observed between the tested samples, C10Z90 showed better BTE with regards to other samples with 14.97% and 37.28% for least and highest BP points respectively while the YF50 reference sample exhibited a 10.12% and 27.74% for same referenced points respectively. This differences possibly resulted from YF50 having no alcohol-based fuel and at 50 ppm of the nano-additive in B20 (which constitute YF50) although aids atomization to create better burning but showed increased ignition delay compared to those with both additives, this agrees with findings in [36–38] that reported higher BTE for all load conditions in samples with alcohol-based additive compared to diesel-biodiesel only.

#### 4.3.2. Brake Specific Fuel Consumption (BSFC)

Generally, all internal combustion engines either gasoline or diesel engines can only generate power after fuel combustion however, the physico-chemical properties especially the density and heating value of the fuel influences the BSFC which represent fuel amount necessary to generate unit of power output and as such BSFC can be scientifically stated to be directly proportional to 'fuel energy content'. As presented in Figure 9, its obvious from the observed trend of the fuel samples tested to have depicted a reduction in BSFC as engine load correspondingly increases with brake power increase. This is due to the promotion of atomization process and more efficient mixing of air fuel ratio (AFR) as the in-cylinder temperature increases and higher turbulence is experienced. In addition, the BSFC is higher with samples with increased viscosities but lower heating values to complement for achieving correlated power output [39]. This is as observed in Table 5 with respect to C10Y90 and C10Z90 in which case viscosity and heating values are 4.09 mm²/s and 45.86 MJ/kg with 4.09 mm²/s and 46.55 MJ/kg respectively, this therefore agrees when extended to observations on Figure 10 whereby at highest BP point the BSFC of C10Z90 is 242.27 g/Kw.hr and C10Y90 is 250.56 g/Kw.hr for same point.

In comparison to YF50 with BSFC of 326.79 g/Kw.hr for same point, it is an indication that addition of butanol further decreases the BSFC making all the samples that are enriched with the butanol in conjunction with the $Fe_3O_4$ nano-additive to perform better than YF50 which have their BSFC values at highest BP point to be 260.50 g/Kw.hr and 282.05 g/Kw.hr for A5Y95 and A5Z95 respectively. Similarly, at the lowest BP point the samples with both $Fe_3O_4$ nano-additive plus butanol still showed lower BSFC than that with only $Fe_3O_4$ nano-additive [3,40,41].

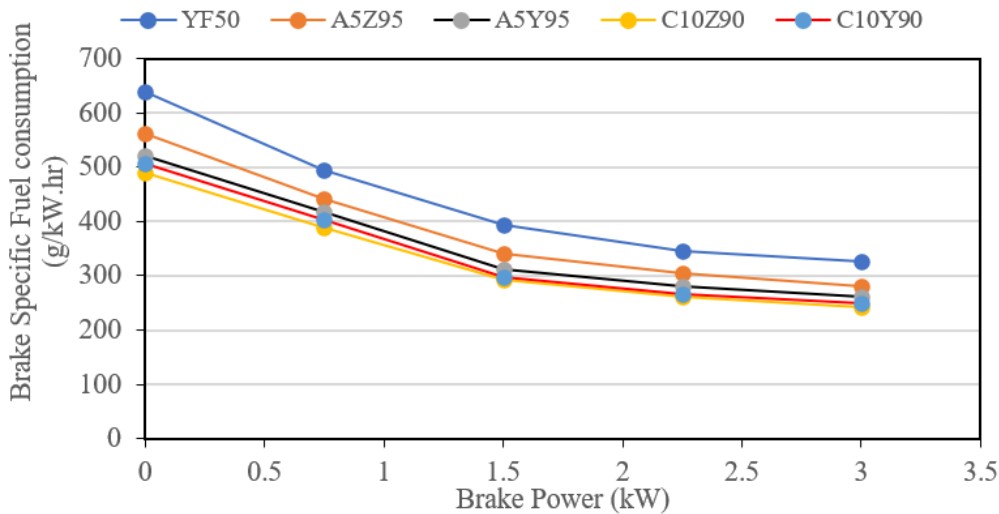

**Figure 9.** BSFC variation with BP.

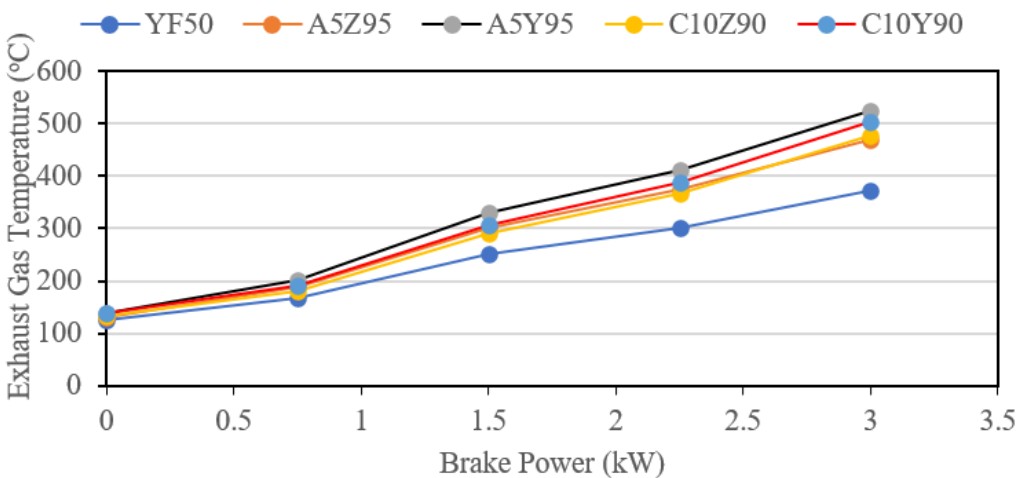

**Figure 10.** EGT Variation with BP.

It therefore indicated that better speed of flame and lower quenching distance associated with butanol in addition to its energy content helps in improving combustion rate and in turn decreases the BSFC just as the magnetite nano-additive also tends to influence this process through their catalytic enhancement of the fuel and thus, showing samples with dual additive to exhibit a lower BSFC at all load conditions compared to those without additives [20,40,42–44].

### 4.3.3. Exhaust Gas Temperature (EGT)

As presented in Figure 10, the EGT increases with corresponding increase in BP for all the fuel samples investigated. The EGT which denotes the energy amount lost as exhaust heat released. The rise in EGT with corresponding higher BP is therefore attributed to higher temperature at higher BP, normally, Y (B20) is reported to exhibit higher EGT than pure diesel fuel [45], however, the introduction of 50 ppm $Fe_3O_4$ nano-additive in Y (B20) tends to diminish the EGT because more heat is released inside the combustion chamber thereby showing YF50 (372.95 °C at highest BP) to have lowest EGT compared to the other samples A5Y95 (523.33 °C), C10Y90 (503.40 °C), C10Z90 (478.03 °C) and A5Z95 (468.97 °C) at same referenced BP. The EGT is increased with the nano-additive enriched butanol with the POME-Diesel fuel, a considerable reason is that rapid combustion influenced by the presence of alcohol-based fuel (butanol) caused some of the unburnt metal oxide nanoparticles to carry heat after combustion to the exhaust region before settling as soot droplets with the emitted gas carrying part of this heat transferred. This possibility originates from similar findings in [20] whereby after combustion, the soot emitted and collected was analyzed by SEM and noticed to entails same shape as the nanoparticles that was doped in the ethanol fuel blend with biodiesel. As also reported in [45], the heat release rate influenced by the presence of rapidly burning fuel (hydrogen) caused higher EGT in samples investigated by the authors.

### 4.4. Emission Parameters

Emission of gases occurs due to incomplete or partial combustion of fuels used for diesel or gasoline engines. The level of emission is dependent on factors like the fuel type and its physico-chemical characteristics [46]. Addition of fuel additives which alter the physico-chemical nature of the fuel therefore changes the emission pattern of the diesel engine.

### 4.4.1. Carbon Monoxide Emission (CO)

The emission of carbon monoxide originates from lack of enough oxygen presence in the combustion chamber, non-optimal richness in fuel mixture and incomplete

combustion [47,48]. The fuel samples investigated in this work all shows an upward trend as the load in correspondence with the BP is increased as observed in Figure 11. These variations with respect to BP showed YF50 exhibiting higher emission in CO compared to the other tested samples. The CO values for YF50 at highest BP point is 2.083% whereas for samples A5Z95, A5Y95, C10Z90 and C10Y90 are respectively 1.296%, 0.897%, 0.508% and 0.767%. This significant reduction in CO emission as the percentage by mass fraction of the dual additive is increased can be attributed to better combustion efficiency due to the presence of the oxygenated metallic nano-additive which could have possibly raised the percentage of oxygen molecules in addition to the presence of the alcohol class butanol which creates a uniform burning of fuel in the combustion chamber by aiding spray and atomization.

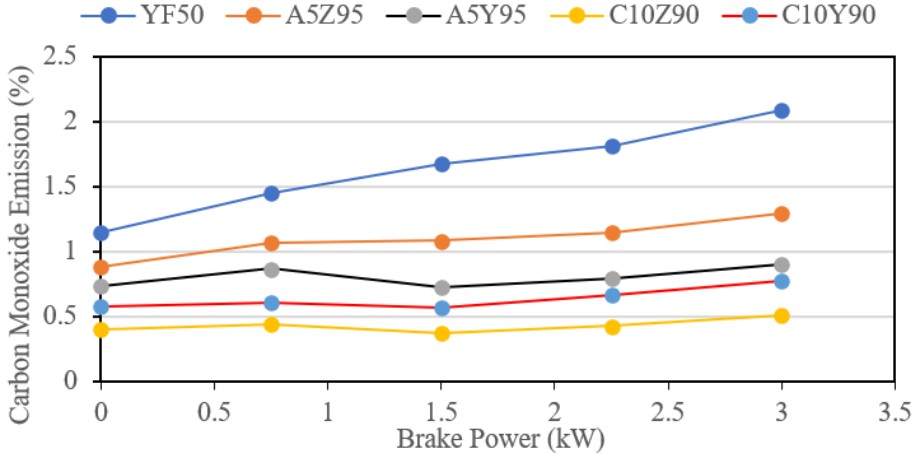

**Figure 11.** CO Emission variation with BP.

The hydroxyl molecule (OH) in butanol will possibly react with more oxygen trapped in air and aids combustion development to cause more reduction in CO formation. Additionally, $Fe_3O_4$ nano-additive contributes in converting the CO molecules into carbon dioxide ($CO_2$) [47], furthermore, biodiesel percentage by ratio in C10Z90 is higher compared to the other samples so expectedly should burn cleaner than the other investigated samples plus $Fe_3O_4$ nano-additive causes turbulence which in turn reduces the rich mixture formation in the combustion chamber to reduce the CO emission as similarly reported by [20,49].

4.4.2. Hydrocarbon Emission (HC)

Formation of unburnt hydrocarbon is based on the incomplete combustion of the fuel in the combustion chamber. As observed from the Figure 12, the variations in hydrocarbon emission with respect to BP for all the tested fuel at various load indicated a rise in HC emission corresponding with the increase in BP. This occurrence possibly resulted from excessive injection of fuel in conjunction with poor or low evaporation rate of the droplets with regards to the engine load and thus, allowing HC to build up over time [50]. The samples investigated showed YF50 fuel exhibiting higher HC emission with its HC value at highest BP point to be 42.31 ppm compared to other samples whereas A5Z95, A5Y95, C10Y90 and C10Z90 having HC emission values at reference point to be 38.54 ppm, 37.36 ppm, 34.45 ppm and 27.88 ppm respectively. The inclusion of $Fe_3O_4$ nano-additive tends to creates higher catalytic activity through elevation of the energy produced in the cylinder of the engine due to ratio of surface area to volume thereby assisting the combustion process and in turn reducing the emission of HC. It is notable that for the samples enriched with the butanol as a co-additive in conjunction with $Fe_3O_4$ nano-additive, the HC emission was lower for such samples, this has caused a difference of up to

14.43 ppm in HC value between YF50 and C10Z90 representing samples with highest HC emission and lowest HC emission at highest BP reference point.

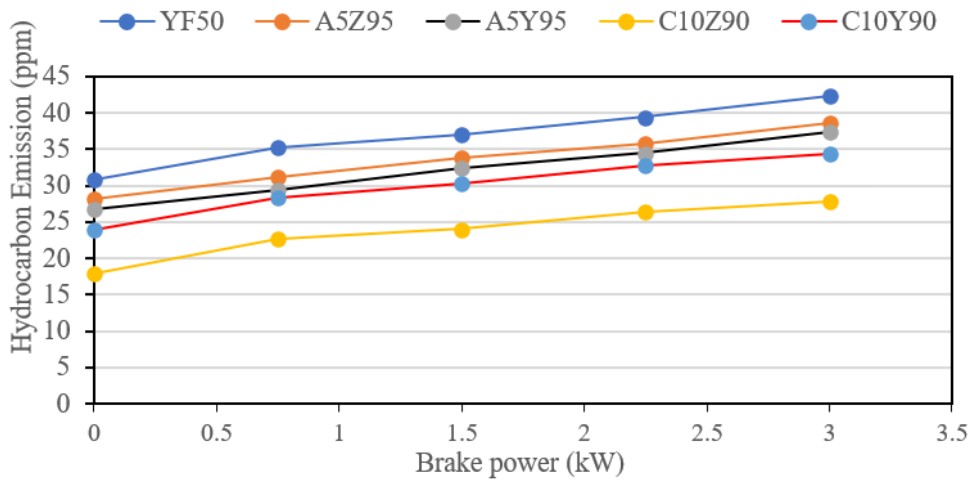

**Figure 12.** HC Emission variation with BP.

It is also relevant to note that the difference of 6.56 ppm of HC emission observed between C10Y90 and C10Z90 at highest BP point could be owning to cleaner burning potential of diesel fuel as the biodiesel content increased such that the presence of butanol and $Fe_3O_4$ nano-additive despite been equal in both samples yet the initial biodiesel ratio in Z (B30) must have contributed to this outcome as Y (B20) have lower biodiesel content similar to findings in [51].

### 4.4.3. Smoke Emission

The emergence of smoke is associated with hefty mass of fuel and therefore occurs frequently at peak loads spontaneously. As observed from the Figure 13 the smoke emission is depicted to rise with corresponding increment in the load conditions and thus in BP such that at highest BP point YF50 exhibited smoke emission level of 49.44 HSU while A5Z95, A5Y95, C10Z90 and C10Y90 showed smoke level of 44.15 HSU, 37.26 HSU, 32.63 HSU and 35.28 HSU respectively for the highest BP points.

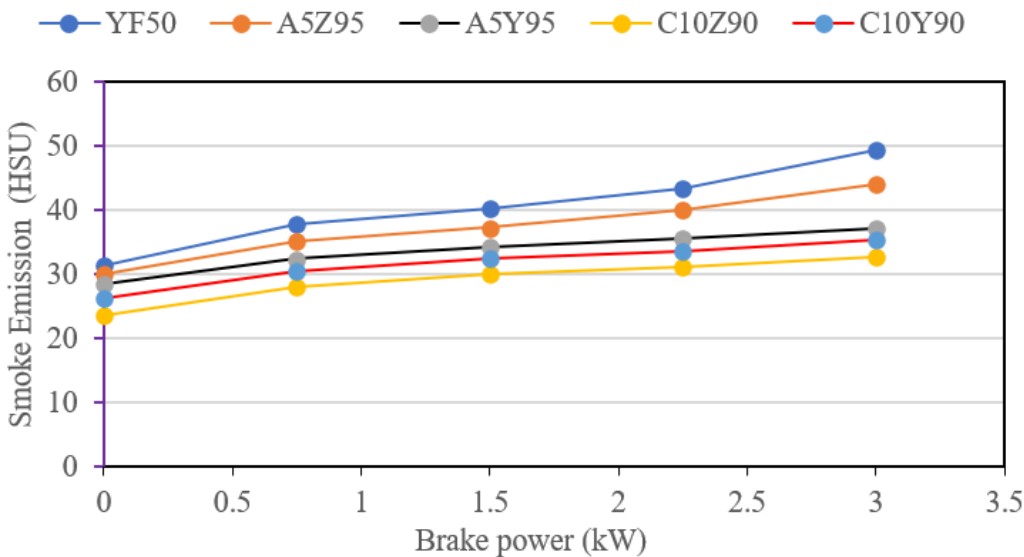

**Figure 13.** Smoke Emission variation with BP.

The lower level of smoke emission noticed with C10Z90 in Figure 13 resulted from inbuilt oxygen presence base on the biodiesel percentage in the blend as well as supporting combustion process which is aided by the butanol content to further instigate the combustion of the HC molecules in the fuel sample [27]. Additionally, application of $Fe_3O_4$ nano-additive is noticed to affect the smoke emission level to decrease especially with samples categories C10Z90 and C10Y90 with dosage of 150 ppm $Fe_3O_4$ nano-additive with butanol observed to emit lower smoke to those with 75 ppm dosage. This of course points out the fact that the $Fe_3O_4$ nano-additive tends to donates more oxygen to aid the combustion process in addition to its high level of thermal conductivity that might have influenced carbonaceous particle oxidation through conveyance of heat to each combustion charge unit on the surface of the $Fe_3O_4$ nano-additive [52,53].

Additionally, butanol present in the present in another way decreases the smoke emission by creating a secondary oxidation sequence in the combustion process in which case the unburnt HC in the cylinder and soot particles can be re-combusted [54].

### 4.4.4. Nitrogen Emission (NOx)

Nitrogen emission in diesel engines results from elongated combustion duration, extreme flame temperature in the combustion chamber and surplus oxygen impendence leading to reaction with nitrogen chemically [55,56], the Figure 14 shows the relation with emission of nitrogen with respect to the increasing load and corresponding increasing BP. It is obvious that for all the tested fuel sample the NOx emission increases as the BP increases. YF50 showed the lowest in terms of NOx emission compared with other samples with its values at highest and second highest BP points to be 796 ppm and 573 ppm, with the addition of nano-additive, NOx was low however, it is seen to increase with the other samples especially A5Y95 which exhibited the highest among the tested fuel with highest and second highest BP point having NOx values of 1141 ppm and 943 ppm, this can be due to higher fuel mass flow rate in the combustion chamber as well as increased concentration of oxygen arising from further oxygen carrying additive butanol which is imbedded into the other samples causing more reaction with nitrogen therefore samples A5Z95, A5Y95, C10Z90 and C10Y90 have NOx values at highest BP and second Highest BP point to be respectively 1036 ppm and 830 ppm, 1141 ppm and 943 ppm, 1061 ppm and 851 ppm, 1095 ppm and 893 ppm.

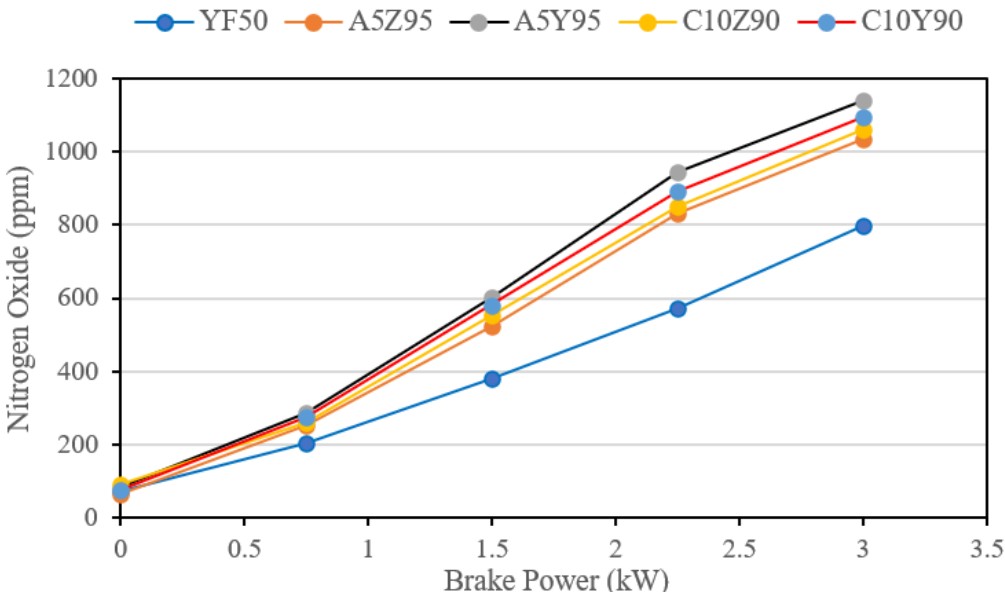

**Figure 14.** Nitrogen Oxides Emission variation with BP.

In addition, tested samples with more percentage ratio of biodiesel tends to have higher NOx as more hydrocarbon chain with oxygen are broken in the combustion process thus, the inclusion of the nano-additive act as booster in this breaking effect through catalytic enhancement in actuating a superior rate of combustion which in turn reduces NOx emission [50].

### 4.4.5. Artificial Neural Network (ANN) Predictions

The architectural design of the ANN model is as presented in Figure 15, As explained in Section 3, the ANN model was formulated and evaluated using the Equations (4) and (5). The training, testing and validation of the developed model showed R value to be below 1 and thus the ANN simulation is accurate in the predictions targeted. In addition, statistical error estimation tool celled Mean Absolute Percentage Error (MAPE) was used to verify and validate the ANN accuracy. These results are as presented in Table 6. The MAPE is as defined in Equation (6).

$$\text{MAPE} = \left\{ \frac{100}{n} \sum_{i=1}^{n} \left| \left( \frac{M_i - P_i}{M_i} \right) \right| \right\} \% \tag{6}$$

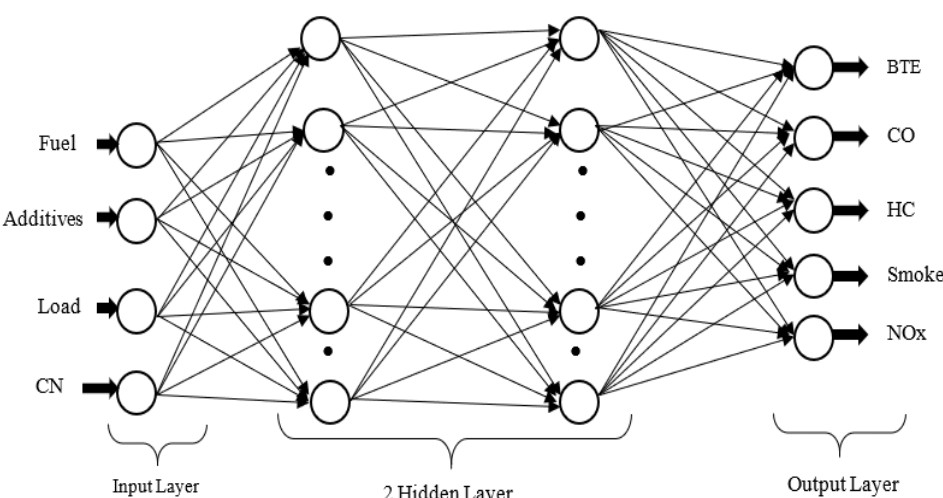

**Figure 15.** ANN Architecture.

The BTE, CO, HC, smoke and NOx was evaluated in varying test cases for experimental and ANN prediction comparison, additionally, the program was run twice to authenticate it for each test case.

**Table 6.** Performance and Error of the Prediction Model.

|  | BTE | CO | HC | Smoke | NOx |
|---|---|---|---|---|---|
| R | 0.998 | 0.997 | 0.999 | 0.997 | 0.989 |
| $R^2$ | 0.997 | 0.994 | 0.999 | 0.993 | 0.977 |
| MAPE (%) | 0.094 | 0.024 | 0.16 | 0.010 | 0.667 |

The result shows that the MAPE values for BTE, CO, HC, smoke and NOx ranges from 0.010 to 0.667, since this value are quite low and that of R and $R^2$ are close to 1, the proposed model is therefore applicable in predicting the targeted output with minimal and acceptable error range. NOx has the highest MAPE value at 0.667 while smoke value was lowest at 0.010.

### 4.4.6. Tested Samples Comparison with Reference Fuel Sample $B_n10Y90$

The best sample in terms of BTE and also in terms of each gas emission was compared with $B_n10Y90$ fuel which was tested under same specific condition with the samples. The results are presented in Figure 16a–e. With respect to BTE, the best tested sample was C10Z90 which is also best sample with minimum emissions for CO, HC, smoke while A5Z95 was best for NOx with YF50 exemption and thus these were used for the comparison with $B_n10Y90$.

It is obvious that in case of BTE presented in Figure 16a, using nano-additive in conjunction with butanol as dual additive to POME and diesel fuel yielded better BTE than a single of either of the additive usage. At the maximum BP tested the BTE for fuel blend $B_n10Y90$ was 23.29%, YF50 was 27.74% whereas C10Z90 which entails both additive exhibited BTE value of 37.28% out performing YF50 with about 10% and $B_n10Y90$ with approximately 14%.

In comparison of emission pattern between usage of butanol or $Fe_3O_4$ nano-additive singly in POME-diesel blends and on the other hand with respect to a combination of both additive to POME-diesel blend, the result is thus presented in Figure 16b–e. It is clear that emission of CO at lowest and highest BP for $B_n10Y90$, YF50 and C10Z90 are respectively 0.153% and 1.931%, 0.146% and 1.814%, 0.10% and 0.422% which prove that the fuel sample C10Z90 which entails both additives is lower in CO emission up to 1.509% at highest BP when compared with $B_n10Y90$ and 1.392% when compared with YF50.

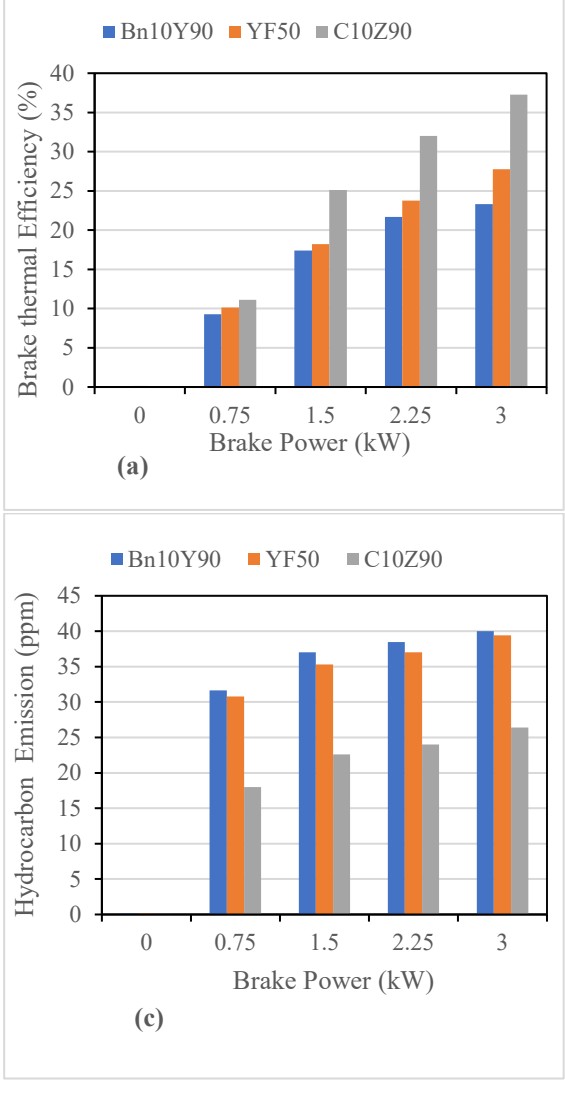

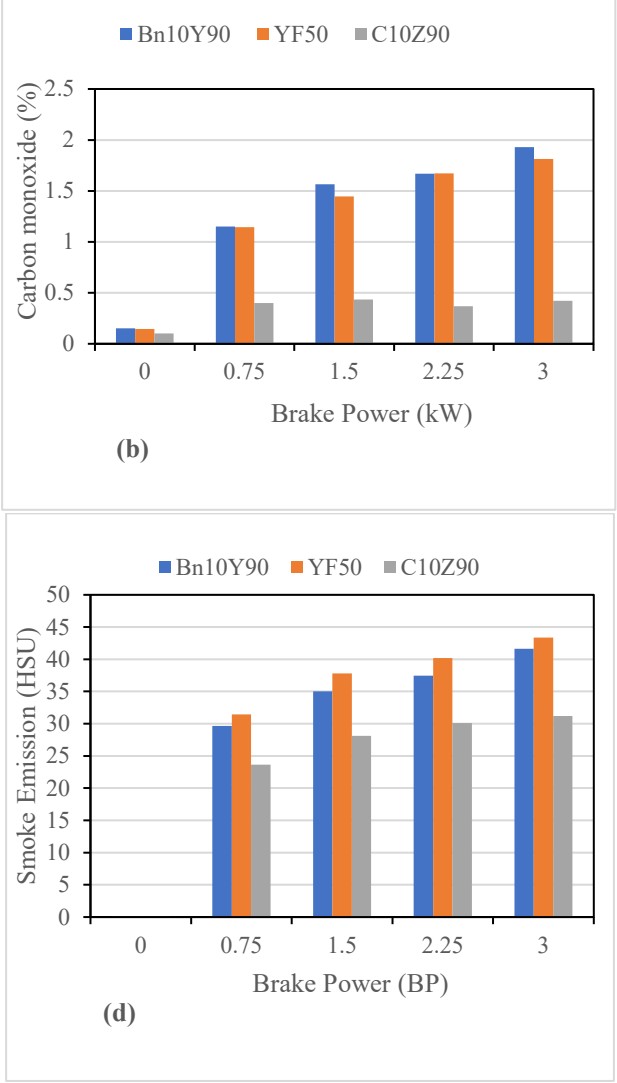

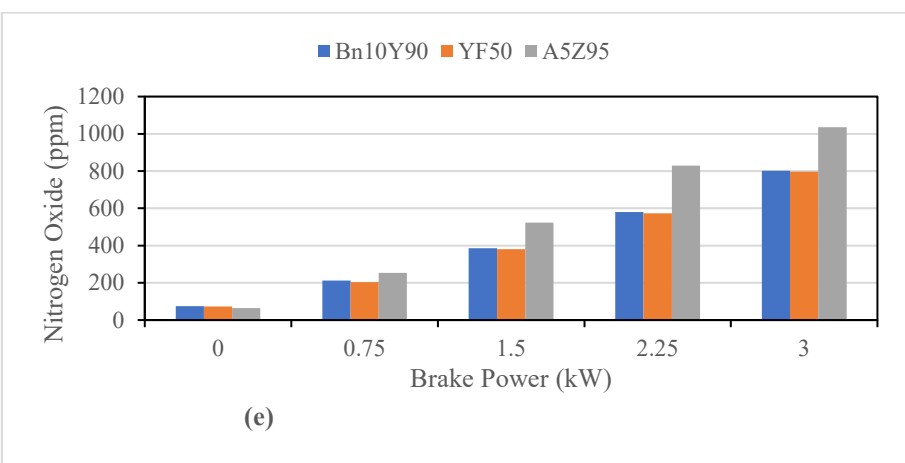

**Figure 16.** (**a**) BTE variation with BP; (**b**) CO Emission variation with BP; (**c**) HC Emission variation with BP; (**d**) Smoke Emission variation with BP; (**e**) Nitrogen Emission variation with BP.

The HC emission of the fuels compared as presented in Figure 16c also further support that combining the two additives yielded better emission pattern of HC as it is lower in tested C10Z90 sample with value at 26.378 ppm at highest BP point while $B_n$10Y90 and YF50 are 40.001 ppm and 39.407 ppm respectively which thus indicated an HC reduction up to 13.623 ppm and 13.029 ppm for each compared case. This of course is attributed to a cleaner and better combustion experience when POME-diesel is doped with the dual additive causing more burning of the fuel sample rather the single of each additive.

In terms of smoke (Figure 16d), higher smoke level was observed with YF50 followed by $B_n$10Y90 and then C10Z90 which showed lowest in smoke emitted. This can be as a result of particles that are left unburnt with the nano-additive percentage composition in YF50 possibly leading to more soot, C10Z90 lower emission of smoke may be due to higher biodiesel content and the combination of butanol to $Fe_3O_4$ nano-additive before blending with B30 could have created a trade-off whereby the butanol causes more rapid combustion of the $Fe_3O_4$ nano-additive in the blend while the $Fe_3O_4$ nano-additive with its energy content causes more breaking of the hydrocarbon chain to combust at a better efficiency. The synergy between the constituent of the C10Z90 sample makes the overall combustion within the combustion chamber quite optimal and in turn leads to overall reduction in smoke emission. The smoke values for C10Z90, YF50 and $B_n$10Y90 at highest BP point are respectively 31.180 HSU, 43.375 HSU and 41.613 HSU. Therefore, the dual additive approach as seen with C10Z90 is lower in smoke emission by 12.195 HSU compared to YF50 and 10.433 HSU compared with $B_n$10Y90.

Emission of NOx is very challenging as it leads to acid rains and devastating health issues, the fuel samples tested in this work showed YF50 having the lowest emission compared to the other tested fuel samples than A5Z95 as observed in Figure 16e. In comparison with the butanol blended reference sample $B_n$10Y90, the emission of NOx is observed higher with A5Z95 which exhibited a value of 1035.95 ppm at highest BP point while YF50 and $B_n$10Y90 at same BP level was noted to be 796.177 ppm and 801.113 ppm sequentially. This definitely have resulted from increased oxygen concentration that is arising from further oxygen carrying additive butanol which is imbedded into the other samples causing more reaction with nitrogen but this is suppressed by the $Fe_3O_4$ nano-additive in the case of YF50 which could have limited the nitrogen formation. YF50 thus is lower in NOx emitted by up to 239.773 ppm compared with $B_n$10Y90 and lower by 234.837 ppm when compared with A5Z95 sample.

## 5. Conclusions

The essence of this work is to understand the impact of dual additive approach of blending $Fe_3O_4$ nano-additive and butanol mixture to POME-diesel fuel and examine

performance and emission pattern, then compare these patterns with use of single additive approach which is already proven to be better than POME-diesel fuel alone as previously investigated by others. Furthermore, ANN was tested to predict results and estimation was made for the model validity using MAPE. With respect to the resulted outcome from the investigations carried out in this study, the following conclusions are identified:

1.  The application of the investigated dual additives to POME-diesel fuel yielded significant improvements with respect to performance and emission levels.
2.  The sample YF50 which implies application of 50 ppm $Fe_3O_4$ nano-additive alone to POME-diesel fuel (B20) yielded lower performance in BTE than all fuel categories that contain both $Fe_3O_4$ nano-additive and butanol.
3.  The addition of the dual additive significantly increased the BTE in all the samples blended with both additive and fuel sample C10Z90 showed best performance in BTE comparatively.
4.  The better performance of C10Z90 over other samples in BTE is based on increased dosage of $Fe_3O_4$ nano-additive and higher biodiesel percentage composition in C10Z90 which is 30% (B30, refer Table 1), by increasing the dosage of $Fe_3O_4$ nano-additive in the initial blend with butanol from 75 ppm to 150 ppm more catalytic inducement for combustion is thus associated with C10Z90.
5.  The doping of the nano-additive firstly in the butanol additive considerably led to higher stability profile of the nano-additive in the final blend leading to slower sedimentation rate and reduction in agglomeration formation.
6.  In relation to emission levels, C10Z90 further showed lower level for CO, HC and smoke but however was higher in NOx than A5Z95. This is quite expected as longer hydrocarbon chain has tendency for higher nitrogen formation at higher temperature in the combustion chamber, YF50 has lowest NOx result since the nano-additive must have acted to be secondary energy carrier and therefore stimulates the fuel particle oxidation and hence leads to lower toxic gases emission.
7.  Considering that the modified fuel C10Z90 has shown remarkable characteristics in improving engine performance and reducing emission level, it is thus recommended as an alternative and sustainable fuel option. Further research on nano-additive should be investigated and commercialized in automobile industries.
8.  The ANN model developed was able to predict the performance and emission pattern of the tested fuels and thus this model fits to be adopted to reduce cost of testing too many varieties of nanoparticles or nanofluids.
9.  The cost of processing nanoparticles is relatively high and may cause higher cost of the final model fuel, this can be modified through determination of alternative production techniques, hence further research is necessitated in this area.
10. The notable increase in smoke level with YF50 makes it necessary for further investigation in respect to how to resolve the issue with unburnt nanoparticles which in a sense can cause other health challenges like cancer and respiratory problems.
11. Nano-additive can be blended with higher viscous fluids and utilized as nano-lubricants due to improved sliding and delamination characteristics of nanoparticle which can improve the lubrication tendencies especially in hydraulic brake systems, application can be extended to nano-coolants due to their heat absorption capability making them feasible in regulating temperature in transformers, ships and nuclear reactors among others.

**Author Contributions:** Conceptualization, A.S.; Methodology, M.E.M.S.; Software, I.V. and M.E.M.S.; Validation, I.V.; Formal analysis, A.S. and M.A.A.; Investigation, A.S. and Z.A.L.; Resources, Z.A.L., M.E.M.S., I.H. and V.E.; Data curation, V.E.; Writing – original draft, A.S.; Visualization, I.V.; Supervision, Z.A.L. and M.A.A.; Project administration, M.E.M.S. and V.E.; Funding acquisition, I.H. All authors have read and agreed to the published version of the manuscript.

**Funding:** This research was funded by Malaysian Palm Oil Board (MPOB) in collaboration with Universiti Teknologi Malaysia (UTM) under grant number R. J130000.7309.405 and the Nigerian Tertiary Education Trust Fund Scholarship.

**Institutional Review Board Statement:** Not applicable.

**Informed Consent Statement:** Not applicable.

**Data Availability Statement:** Not applicable.

**Acknowledgments:** The authors of this research paper sincerely appreciate the financial support by the Malaysian Palm Oil Board (MPOB) in collaboration with Universiti Teknologi Malaysia (UTM) under grant number R. J130000.7309.405 and also thank the Nigerian Tertiary Education Trust Fund (TET Fund) for sponsorship.

**Conflicts of Interest:** Authors of this work categorically state that there is no competing interest in terms of personal or financial inducement which could have affected or influence the outcome of the results and reporting in this work.

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
