# Peer review of "Dual Effects of N-Butanol and Magnetite Nanoparticle to Biodiesel-Diesel Fuel Blends as Additives on Emission Pattern and Performance of a Diesel Engine with ANN Validation"

_sustainability, doi:10.3390/su15021404_

Round 1
Reviewer 1 Report
In this study, the effects of butanol biodiesel Fe3O4 and diesel mixtures were investigated by experimental and ANN methods.
Studies given in the introduction part are insufficient. Studies on Fe3O4 should be added.
The material and method part is sufficient. It would be more appropriate to make a comparison with the standard diesel data among the experimental data. The effect of the added ingredients will be fully revealed.
There are no in-cylinder pressure data expressed in figure 3 in the results section, and these data are required for interpretation of combustion.
Also, no results were seen for the Bn10Y90 fuel mixture mentioned in line 196.
No CO2 emission results were seen in the emission results.
It was seen that it was not included for all fuels in the model estimated with ANN. Can you explain why?
Reviewer 2 Report
This paper investigates the impact of magnetite dispersed in butanol and added to two varied blends of palm biodiesel and diesel B20 and B30. The topic is relevant and fully corresponds to the subject matter of the Energies journal. Some separate structural additions and a number of clarifications of technological aspects of the article material should be made:
- Novelty: The authors partly stated what motivated the idea portrayed in this study, what then does this study offer beyond the recent advances made on the combined approaches? I would advise the authors to carefully carry out a close comparison as well in the discussion section to enumerate the advantages this study offers over other related works.
- Most of the ideas written were already described in many literatures. The Authors tried to compile it but lack of the enhancement of the interrelation analysis between the references. It is advised that the authors give a deeper analysis on how these ideas become more applicative strategies so that they can contribute to the next step of implementation.
- Introduction provides a great overview and introduces the topic. However, it misses the aim of the study or what is going to be done in this study. Please state clearly the aims and what this study does. However, highlight that innovative combustion systems can improve the CO2 and emissions with innovative injection strategies and FIS (SAE 2018-01-1697) in combination with advanced combustion concepts and alternative fuels (10.1080/15567036.2022.2124326; 10.1299/jmsesdm.2017.9.C308). These technologies/fuels could be used also to improve efficiency and performance. The authors could extend the introduction discussion reporting that innovative technologies could give a potential boost to the CI engine fuel economy and engine-out emissions reduction.
- More in-depth analysis of the author's contribution of this paper in the introduction section. I would like to see more discussion of the literature so that I can clearly identify the article relates to competing ideas.
- The motivation of the paper is unclear, while it should be eye catching to make more sense. In this regard, a separate section on motivation and contribution should be included.
- The results of the study of are not to a small extent dependent on the accuracy class of the measuring equipment used. Please complete the methodology section with appropriate information.
- How many repeat experiments were performed at each point? Comment on repeatability.
- The language of the manuscript is fair; I would advise consulting a language editor to further polish the language of the manuscript. There are several grammatical mistakes. Please work close to a native English speaker to refine the language of this paper.
- Challenges and future directions to improve and implement of these technologies with big data analytics should be discussed.
- Further explanation of the advantageous of the suggested approach should be added. What are the main positive and negative points of this approach?
- In my opinion, there are several up-to-date approaches for the idea. Authors should look for these approaches, compare the results and prove their idea. This is the major concern.
- Future scope, and current limitations must be discussed, for example, a short paragraph may be included in the conclusion section more explicitly.
- How did the authors establish/estimate the uncertainties in the measurements documented in this research?
Reviewer 3 Report
The article in question deals with adding iron nanoparticles dispersed in butanol to observe different effects in diesel and biodiesel blends.
The manuscript is well-written and organized. However, some points must be taken into account:
For me, the effect of the presence of nanoparticles is not clear, since all tests should also be done without them.
Another point is that all tests should be done with the fuels separately (Y and Z), before mixing and obtaining the blends.
See how to cite references in the text in this journal. Line 330 "this agrees with findings in [37-39]".
The results should be better compared with the values already existing in the literature. For example: Is this the desired acidity? Are all the values observed in the tests within the expected range?
Round 2
Reviewer 1 Report
The corrections made in the study and the answers given to the referees are sufficient.
Reviewer 2 Report
The authors have improved the manuscript.